# Intranasal Administration of Recombinant Newcastle Disease Virus Expressing SARS-CoV-2 Spike Protein Protects hACE2 TG Mice against Lethal SARS-CoV-2 Infection

**DOI:** 10.3390/vaccines12080921

**Published:** 2024-08-16

**Authors:** Deok-Hwan Kim, Jiho Lee, Da-Ye Lee, Seung-Hun Lee, Jei-Hyun Jeong, Ji-Yun Kim, Jiwon Kim, Yang-Kyu Choi, Joong-Bok Lee, Seung-Young Park, In-Soo Choi, Sang-Won Lee, Sungsu Youk, Chang-Seon Song

**Affiliations:** 1Avian Disease Laboratory, College of Veterinary Medicine, Konkuk University, Seoul 05029, Republic of Korea; ejrghks41@gmail.com (D.-H.K.);; 2KHAV Co., Ltd., 1 Hwayang-dong, Gwangjin-gu, Seoul 05029, Republic of Korea; 3Southeast Poultry Research Laboratory, U.S. National Poultry Research Center, U.S. Department of Agriculture-Agricultural Research Service, 934 College Station Road, Athens, GA 30605, USA; 4Department of Microbiology, College of Medicine, Chungbuk National University, Cheongju 28160, Republic of Korea; 5Department of Laboratory Animal Medicine, College of Veterinary Medicine, Konkuk University, Seoul 05029, Republic of Korea; yangkyuc@konkuk.ac.kr; 6Biomedical Research Institute, Chungbuk National University Hospital, Cheongju 28644, Republic of Korea

**Keywords:** SARS-CoV-2, Newcastle disease virus vector-based vaccine, intranasal vaccine, lung viral load, SARS-CoV-2 spike-specific IgA

## Abstract

Coronavirus disease 2019 (COVID-19), caused by severe acute respiratory syndrome coronavirus-2 (SARS-CoV-2), emerged as a global outbreak in 2019, profoundly affecting both human health and the global economy. Various vaccine modalities were developed and commercialized to overcome this challenge, including inactivated vaccines, mRNA vaccines, adenovirus vector-based vaccines, and subunit vaccines. While intramuscular vaccines induce high IgG levels, they often fail to stimulate significant mucosal immunity in the respiratory system. We employed the Newcastle disease virus (NDV) vector expressing the spike protein of the SARS-CoV-2 Beta variant (rK148/beta-S), and evaluated the efficacy of intranasal vaccination with rK148/beta-S in K18-hACE2 transgenic mice. Intranasal vaccination with a low dose (10^6.0^ EID_50_) resulted in an 86% survival rate after challenge with the SARS-CoV-2 Beta variant. Administration at a high dose (10^7.0^ EID_50_) led to a reduction in lung viral load and 100% survival against the SARS-CoV-2 Beta and Delta variants. A high level of the SARS-CoV-2 spike-specific IgA was also induced in vaccinated mice lungs following the SARS-CoV-2 challenge. Our findings suggest that rK148/beta-S holds promise as an intranasal vaccine candidate that effectively induces mucosal immunity against SARS-CoV-2.

## 1. Introduction

Coronavirus disease-19 (COVID-19), caused by severe acute respiratory syndrome-coronavirus-2 (SARS-CoV-2), spread globally from late 2019 to early 2020 and was officially declared a pandemic by the World Health Organization (WHO) in March 2020 [1]. The primary symptoms of COVID-19 range from asymptomatic to severe pneumonia, potentially leading to ICU admission or death [2]. On 11 May 2023, the WHO officially declared that COVID-19 was no longer a global public health emergency. According to the WHO’s Coronavirus Dashboard, approximately 772 million people have been infected with SARS-CoV-2, with around 7 million deaths recorded by 30 November 2023 [3]. Injectable vaccines, including mRNA vaccines, adenovirus vector-based vaccines, subunit vaccines, and inactivated whole-virion (WV) vaccines targeting SARS-CoV-2, have been developed and used globally to combat COVID-19 [4,5,6,7,8]. Despite vaccine successes, SARS-CoV-2 continues to spread, causing infections and fatalities, with ongoing concerns about the emergence of new variants.

The tonsils and adenoids, known as the nasal-associated lymphoid tissues, serve as the primary inductive sites for the secretory immune system [9] and are implicated as critical sites for coronavirus infection [10]. Multiple studies have shown that protecting the lungs from viral pathogens can be achieved through the nasal administration of viral and bacterial vaccines [11,12]. Considering that the respiratory lining acts as a gateway for respiratory viruses, the rationale for inducing mucosal immunity via appropriate vaccines also applies to SARS-CoV-2 vaccines. While most SARS-CoV-2 vaccines are administered parenterally via intramuscular injection, inducing the systemic production of virus-specific IgG, the induction of virus-specific secretory IgA in the upper respiratory tract remains modest [7,13]. Although current intramuscular vaccines are highly effective in preventing severe COVID-19, their efficacy in preventing infection and transmission among individuals remains low [14]. Consequently, there is a continued emphasis on the need for vaccines capable of inducing mucosal immunity specific to SARS-CoV-2 [15,16], with numerous studies currently investigating the elicitation of mucosal immunity using such vaccines [17,18,19].

Newcastle disease virus (NDV) is a non-segmented, single-stranded, negative-sense RNA virus with a genome size of 15.2 kb, belonging to the family *Paramyxoviridae* and the genus *Orthoavulavirus*. NDV comprises the following six structural proteins: nucleoprotein (NP), phosphoprotein (P), matrix protein (M), fusion protein (F), hemagglutinin–neuraminidase (HN), and large protein (L) [20]. Previous research has indicated that NDV can reliably accommodate foreign genes up to 5 kb [21,22], with no discernible compromise in its ability to propagate within eggs and cells [23]. Although NDVs are prevalent in various avian species, including domesticated poultry and wild birds [24], human exposure to NDVs remains low and only rarely results in self-resolving conjunctivitis in poultry workers [25]. Additionally, NDV is considered safe as an oncolytic virus in humans, even with high doses of live viruses administered clinically [26]. The NDV vector has been used as the platform for the following vaccines: (1) SARS-CoV-2; the SARS-CoV-2 vaccine is also applicable for use in pigs, and clinical trials in humans have already been completed [27,28], (2) the bovine ephemeral fever virus vaccine for used bovine [29], and (3) the rabies virus vaccine for use in canines and felines [30].

To leverage the advantages of NDVs as viral vectors, we utilized a heat-resistant NDV strain (K148/08) [31] and developed a SARS-CoV-2 vaccine (rK148/beta-S). In a previous study, intramuscular vaccination with rK148/beta-S protected against the SARS-CoV-2 Beta and Delta variants [32]. In this study, we investigated the potential application of rK148/beta-S as an intranasal live vaccine in K148-hACE2 transgenic mice. We assessed both humoral and cellular immune responses and subsequently evaluated protection against challenges with SARS-CoV-2 Beta or Delta variants.

## 2. Material and Methods

### 2.1. Virus and Cells

HEp-2 (CCL-81; American Type Culture Collection, Manassas, VA, USA) and Vero-E6 (CRL-1586; American Type Culture Collection) cell lines were maintained in Dulbecco’s modified Eagle’s medium (DMEM) supplemented with 8% fetal bovine serum (FBS; Biowest, Nuaillé, France) and antibiotics, at 37 °C in a 5% CO_2_ incubator. SARS-CoV-2 Beta and Delta variants were sourced from the Korea Disease Control and Prevention Agency. Propagation of SARS-CoV-2 was carried out using the Vero-E6 cell line. The recombinant NDV-vectored SARS-CoV-2 vaccine (rK148/beta-S) was propagated in 10-day-old specific pathogen-free (SPF) embryonated chicken eggs (ECEs), and the 50% egg infective dose (EID50) of the virus was determined. All experiments involving viable SARS-CoV-2 were conducted in a biosafety level-3 (BSL-3) facility at Konkuk University, adhering to procedures approved by the Konkuk University Institutional Biosafety Committee (approval no. KUIBC-2023-03).

### 2.2. Mouse Immunization and Challenge

Immunization and infection studies involving animals were reviewed, approved, and supervised by the Konkuk University Institutional Animal Care and Use Committee (approval no. KU23053). Female K18-hACE2 mice (six-week-old, *n* = 96) were procured from Jackson Laboratory (Bar Harbor, ME, USA) and acclimated for one week. The vaccine study groups were further divided into six groups. Groups 1–4 were designated for SARS-CoV-2 Beta variant challenge (Figure 1A and Figure 2A), whereas groups 5 and 6 were designated for SARS-CoV-2 Delta variant challenge (Figure 3A). The groups were as follows: G1 (rK148/beta-S; *n* = 7, low-dose), G2 (rK148/beta-S; *n* = 21, high-dose), G3 (NDV K148/08; *n* = 21), G4 (control: phosphate-buffered saline (PBS); *n* = 21), G5 (rK148/Beta; *n* = 13), and G6 (control: PBS; *n* = 13). Intranasal administration of the vaccine or challenge virus was conducted by instillation into the nostrils following anesthesia. Each vaccine was administered intranasally with 100 μL of live virus at a concentration of either 10^7.0^ EID_50_/mL (low dose) or 10^8.0^ EID_50_/mL (high dose) (Table 1). All groups received the secondary vaccine four weeks post-primary vaccination. Blood samples were collected at 4, 6, and 8 weeks post-primary vaccination, and body weight was recorded weekly. Four weeks after the second vaccination, three boost-vaccinated K18-hACE2 mice from each of groups G2 to G4 were euthanized, followed by spleen extraction and ELISpot analyses. Then, the remaining boost-vaccinated K18-hACE2 mice were intranasally inoculated with 50 μL of SARS-CoV-2 Beta or Delta variant (10^6.0^ EID_50_/mL). SARS-CoV-2 infection was performed at the BSL-3 animal facility of Konkuk University. Clinical signs, body weight, and mortality were monitored and measured daily for 14 days post-challenge (dpc). Three mice from each group were euthanized at 3 and 6 dpc. Viral load of SARS–CoV–2, SARS-CoV-2 spike-specific IgA, and vaccinated NDV were measured in the lungs. For the SARS-CoV-2 Beta variant challenge groups (G2, G3, and G4), three mice from each group were euthanized at 5 dpc. Lung, spleen, and small intestine samples were collected for histopathological analysis. 

### 2.3. Serological Analysis

Serum antibody titers against NDV (K148/08) were determined using a standard protocol [19]. To remove nonspecific hemagglutination inhibition (HI) factors, the serum was incubated with a receptor-destroying enzyme (Denka Seiken, Japan) at a 1:3 ratio for 18 h at 37 °C. After inactivation, the samples were diluted two-fold with PBS in 96-well V-bottom plates. Subsequently, four hemagglutination units of K148 inactivated antigen were added to each well and incubated for 40 min at room temperature (20–25 °C). The incubated samples were mixed with equal volumes of 1% turkey red blood cells in PBS. HI titers were reported as reciprocal log2 titers.

### 2.4. Surrogate SARS-CoV-2 Enzyme-Linked Immunosorbent Assay (ELISA)

The presence of antibodies against SARS-CoV-2 in the serum was assessed using a surrogate SARS-CoV-2 ELISA kit (BioNote, Hwaseong, Republic of Korea), following the manufacturer’s instructions. The serum obtained from mice was mixed with PBS and diluted 1:10 for use as samples in the ELISA. To measure the surrogate virus neutralization test (sVNT) values for the SARS-CoV-2 Beta variant, a working enzyme conjugated with SARS-CoV-2 Beta and Gamma variants was used [32]. The percent inhibition (PI) value was calculated using the following formula: OD ((1−(sample OD/negative OD)) × 100).

### 2.5. Splenocyte and Interferon-Gamma (IFN-γ) Enzyme-Linked Immunospot (ELISpot)

Mouse spleens were isolated and placed into Roswell Park Memorial Institute (RPMI) 1640 medium (Sigma Aldrich, St. Louis, MO, USA) supplemented with 10% fetal bovine serum (FBS). Single-cell suspensions were prepared by mincing the spleens using a 70 μm cell strainer (SPL Life Sciences, Pocheon-si, Republic of Korea) in a Petri dish. The mouse IFN-γ ELISpot assay was conducted using a commercial kit (ELISpot Plus/Mouse IFN-γ; Mabtech, Sweden) and PepTivator SARS-CoV-2 Prot_S1 (Miltenyi Biotec, North Rhine-Westphalia, Germany) as antigen. The quantification of spots was performed using the AID iSpot system (AID Autoimmun Diagnostika GmbH, Strasberg, Germany).

### 2.6. Lung Viral Load and Specific IgA against SARS-CoV-2

The lungs were homogenized using a mortar and pestle, and the resulting material was preserved in 10% (*w*/*v*) PBS. After centrifugation at 3000 rpm for 10 min, the supernatant was collected to measure the viral load and SARS-CoV-2 spike-specific IgA. Vero-E6 cell monolayers were inoculated with 25 μL of the supernatant (diluted 10-fold in DMEM). The inoculated cells were incubated at 37 °C with 5% CO_2_ for 40 min, followed by overlaying with 175 μL of 2% FBS in DMEM. The cytopathic effect (CPE) was observed at four days post-inoculation, and the 50% tissue culture infectious dose (TCID_50_) was calculated [33]. The supernatant was analyzed using the Mouse Anti-2019 nCoV(S) IgA ELISA kit (FineTest, Wuhan, China) according to the manufacturer’s protocol. Briefly, 50 μL of either the standard or sample was dispensed into each well and incubated for 30 min at 37 °C. Following a wash step, 50 μL of HRP-labeled antibody working solution was added to each well and incubated for an additional 30 min at 37 °C. After a second wash, 50 μL of 3,3′,5,5′–tetramethylbenzidine (TMB) substrate solution was added and incubated for 20 min at 37 °C. The reaction was then stopped by the addition of 50 μL of stop solution. Absorbance was read at 450 nm, and the viral load and SARS-CoV-2-specific IgA in the lungs were calculated.

### 2.7. NDV Detection from Egg Inoculation and Real-Time Reverse Transcription PCR

The lung samples collected at 3 and 6 dpc were used to assess NDV clearance. The supernatant was inoculated into 10-day-old SPF ECEs and incubated at 37 °C for 3 days. Allantoic fluid was used in the hemagglutination assay to detect the presence of rK148/beta-S or NDV. Total RNA was extracted from the supernatant using the RNeasy kit (Qiagen, Hilden, Germany). NDV RNA levels were quantified by measuring the cycle threshold (Ct) values through real-time reverse transcription PCR targeting the matrix gene [34].

### 2.8. Histopathological Evaluation

The lungs, spleen, and small intestine were excised from euthanized mice and preserved in 10% neutral-buffered formalin for hematoxylin and eosin staining. Histopathological examination was conducted following a standardized protocol as previously described [35]. 

### 2.9. Statistical Analysis

GraphPad Prism 8.0 (Boston, MA, USA; www.graphpad.com (accessed on 6 July 2023) was employed for statistical analyses. The differences between mean titers were statistically evaluated using one-way ANOVA followed by Tukey’s post hoc test for multiple comparisons. A one-tailed *t*-test was used for the comparison between the two groups, and a log-rank test was used for the survival rate analysis. A significance level of *p* < 0.05 was considered statistically significant (* *p* < 0.05, ** *p* < 0.01, and *** *p* < 0.001).

## 3. Results

### 3.1. Immune Response to Intranasal Vaccination of Live rK148/Beta-S

Body weight changes were monitored up to eight weeks post-vaccination (wpv). However, there were no significant differences in body weight among the groups throughout the immunization period (Figure 1B). Four weeks after the initial vaccination, detectable levels of serum HI antibodies against the vector virus were observed in a few mice in the high-dose groups (G2 and G3; Figure 1C). Following booster vaccination, serum anti-Newcastle disease virus (NDV) antibodies were detectable across all immunized groups, with higher titers observed in the high-dose groups (G2 and G3) than in the low-dose group (G1). SARS-CoV-2 enzyme-linked immunosorbent assay (ELISA) results indicated a positive serum antibody titer against the spike protein in a subset of mice administered rK148/beta-S across both low- and high-dose groups at four weeks after the initial vaccination. Following booster vaccination, all mice administered rK148/beta-S (G1 and G2) tested seropositive for SARS-CoV-2, whereas no positive results were observed in the NDV-vaccinated group (G3) or the negative control group (G4) (Figure 1D). Serum antibody analyses indicated that the intranasal administration of rK148/beta-S elicited systemic humoral immunity against SARS-CoV-2 following booster vaccination.

**Figure 1 vaccines-12-00921-f001:**
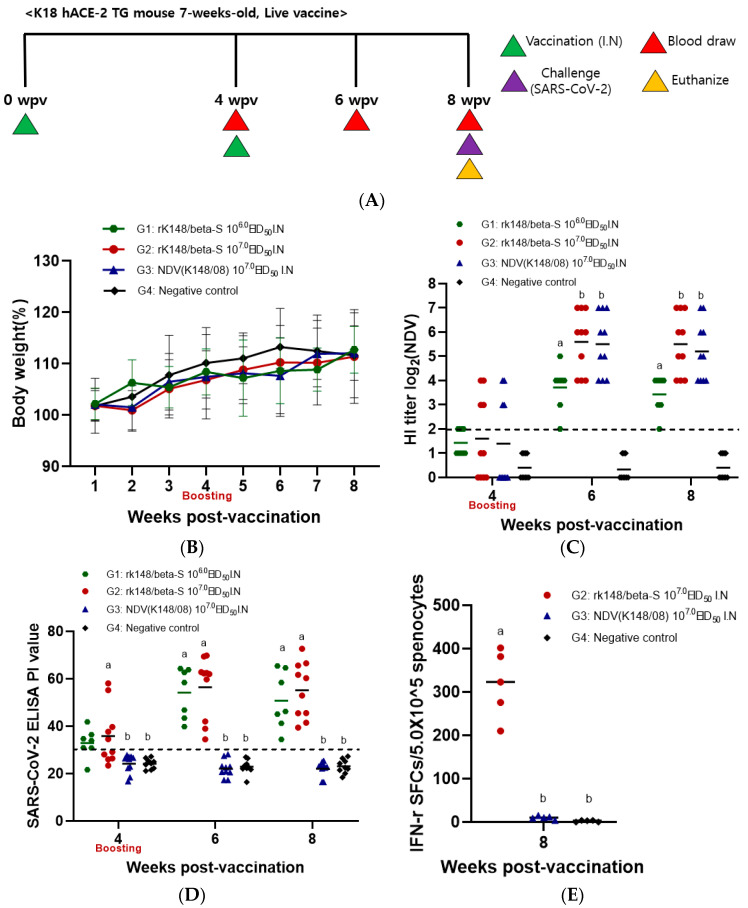
Immune response to intranasal vaccination of rK148/beta-S. (**A**) The schedule for vaccination, blood sampling, and euthanasia is outlined. (**B**) Body weight changes were recorded weekly up to eight weeks post-vaccination (wpv). (**C**) Hemagglutination inhibition (HI) assay was performed on serum samples from vaccinated mice; mice with HI titers < 2 log2 were considered seronegative. (**D**) Surrogate SARS-CoV-2 enzyme-linked immunosorbent assay (ELISA) was conducted on serum samples from vaccinated mice, with serum diluted to 1/10. Mice with surrogate SARS-CoV-2 positivity index (PI) values < 30 were deemed seronegative. (**E**) Splenocyte analysis and IFN-γ enzyme-linked immunospot assays were performed on autopsied animals (*n* = 3 per group) at 4 weeks post-booster vaccination, using PepTivator SARS-CoV-2 Prot_S1 as the antigen. Groups with at least one shared letter superscript indicate no significant statistical differences were observed between pairwise comparisons within the same week (*p* > 0.05).

### 3.2. Efficacy of Intranasal Vaccination with rK148/Beta-S in Protecting Mice against Both SARS-CoV-2 Beta and Delta Variants

To assess the efficacy of rK148/beta-S, mice were challenged with either SARS-CoV-2 Beta or Delta variants at 8 weeks post-vaccination (wpv), which corresponds to 4 weeks post-booster vaccination. Changes in body weight and survival rate were monitored until 14 dpc (Figure 2 and Figure 3A). Following infection with the SARS-CoV-2 Beta variant, the recombinant NDV group (G3) and PBS control group (G4) exhibited 100% mortality by 9 dpc. In contrast, the rK148/beta-S low-dose group (G1) showed an 86% survival rate, while the rK148/beta-S high-dose group (G2) demonstrated no decrease in body weight and achieved 100% survival until 14 dpc (Figure 2B,C). Three mice from each group(G2, G3, and G4) were euthanized at 3 and 6 dpc to assess the lung viral load. At 3 dpc, the lung viral load in the rK148/beta-S high-dose group (G2) was significantly lower than that of the NDV-vaccinated group (G3) and the PBS control group (G4). By 6 dpc, no lung viral load was detected in the rK148/beta-S high-dose group (G2), while viruses were still detected in the NDV-vaccinated group (G3) and the PBS control group (G4) (Figure 2D). 

Following the challenge with the SARS-CoV-2 Delta variant, the PBS control group (G6) exhibited 100% mortality at 9 dpc, while the rK148/beta-S high-dose group (G5) showed no decrease in body weight and maintained a 100% survival rate until 14 dpc (Figure 3B,C). Three mice from each group (G5 and G6) were euthanized at 3 and 6 dpc to measure lung viral load. At 3 dpc, the lung viral load in the rK148/beta-S high-dose group (G5) was significantly lower than that in the PBS control group (G6). By 6 dpc, no lung viral load was detected in the rK148/beta-S high-dose group (G5), whereas viruses were detected in the PBS control group (G6) (Figure 3D).

**Figure 2 vaccines-12-00921-f002:**
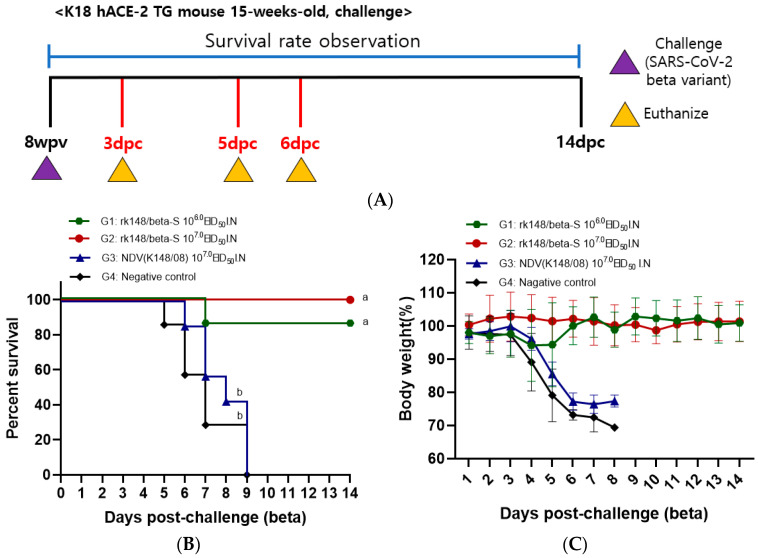
Survival rate, lung viral load, and SARS-CoV-2 spike-specific IgA following challenge with the SARS-CoV-2 Beta variant. (**A**) The schedule for SARS-CoV-2 Beta variant challenge and euthanasia is depicted. (**B**,**C**) Changes in body weight and survival rate were monitored at 14 days post-challenge (dpc) with the SARS-CoV-2 Beta variant. For the challenge, 50 μL of the Beta variant (10^6.0^ EID_50_/mL) was administered intranasally (*n* = 16). (**D**,**E**) Mice were euthanized at 3 and 6 dpc (*n* = 3 per group) to assess viral load in the lungs and measure SARS-CoV-2 spike-specific IgA levels. Groups sharing at least one letter superscript indicate no significant statistical differences between pairwise comparisons within the same week (*p* > 0.05).

**Figure 3 vaccines-12-00921-f003:**
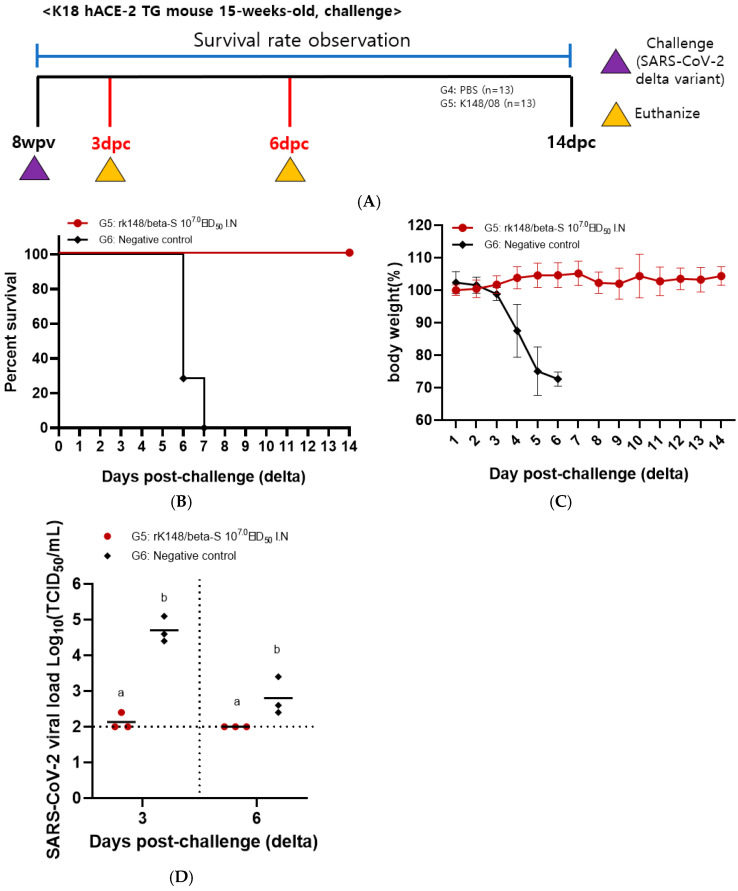
Survival rate and lung viral load following challenge with the SARS-CoV-2 Delta variant. (**A**) The schedule for SARS-CoV-2 Delta variant challenge and euthanasia is outlined. (**B**,**C**) Changes in body weight and survival rate were monitored after challenge with the SARS-CoV-2 Delta variant. For the challenge, 50 μL of the Delta variant (10^6.0^ EID_50_/mL) was administered intranasally (*n* = 13). (**D**) Following the challenge, mice were euthanized at 3 and 6 days post-challenge (dpc) (*n* = 3 per group) to assess viral load in the lungs. Groups with at least one shared letter superscript indicate no significant statistical differences between pairwise comparisons within the same week (*p* > 0.05).

NDV was undetectable in the lung samples collected at 3 and 6 dpc using both real-time PCR and egg inoculation methods. This suggests that the vaccinated viruses were cleared by the time of the challenge.

### 3.3. IFN-γ-Secreting Splenocytes in Response to SARS-CoV-2 and Respiratory Tissue IgA Specific to SARS-CoV-2

Three mice from each of groups G2, G3, and G4 were euthanized at 8 weeks post-initial vaccination. Spleen samples underwent ELISpot assay to measure IFN-γ-secreting immune cells in response to the SARS-CoV-2 spike antigen. SARS-CoV-2-specific IFN-γ spots were detected only in the rK148/beta-S high-dose group (G2; 318.6 ± 70.1) (Figure 1E). To evaluate the SARS-CoV-2 spike-specific IgA in the lungs after SARS-CoV-2 challenge, three mice from each group (G2, G3, and G4) were euthanized, and their lungs were extracted at 3 and 6 days post-challenge. The mean IgA level in the rK148/beta–S high-dose group (G2) was 65.6 (±11.1) ng/mL at 3 dpc and 176.4 (±22.3) ng/mL at 6 dpc. The K148/08 (G3) and PBS control (G4) groups showed SARS-CoV-2 spike-specific IgA levels lower than 10 ng/mL (Figure 2E).

### 3.4. Histopathological Analysis of the Lungs, Spleen, and Small Intestine

To observe the histopathological differences following SARS-CoV-2 Beta variant infection, three mice from each group were euthanized at 5 dpc. The rK148/beta-S- and NDV-vaccinated groups (G2 and G3) showed moderate immune cell infiltration around the bronchioles, with slightly greater infiltration in G2, but maintained overall alveolar structure. In contrast, the control group (G4) demonstrated severe inflammation and disruption of the alveolar architecture, suggesting a lack of protective immunity. No significant differences in the splenic tissue morphology were observed among the three groups. The rK148/beta-S-vaccinated group (G2) showed normal villi structure, whereas the NDV-vaccinated group (G3) and control group (G4) exhibited slightly fewer villi, suggesting possible damage or an inadequate protective response in the absence of vaccination (Figure 4).

**Figure 4 vaccines-12-00921-f004:**
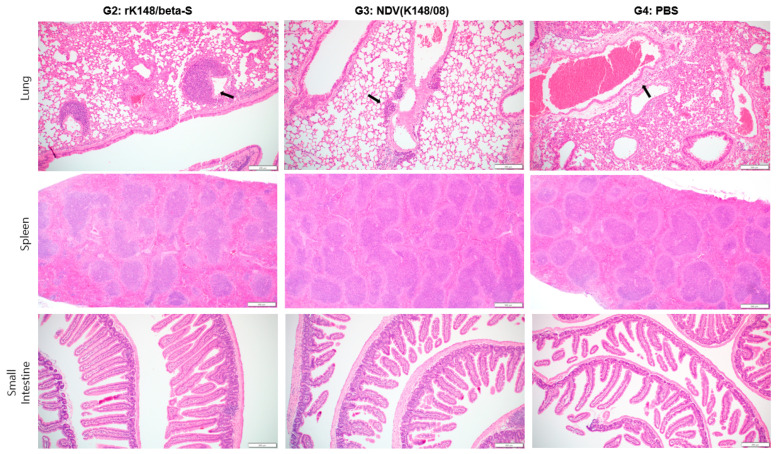
Histopathological analysis of the lung, spleen, and small intestine after challenge with SARS-CoV-2 Beta variant. After the challenge with Beta variant, the mice were euthanized 5 dpc (*n* = 3) to perform histopathological analysis. Black arrows indicate perivascular edema; magnification ×100.

## 4. Discussion

For diseases primarily transmitted through the respiratory mucosa, such as COVID-19 and influenza, intramuscularly administered vaccines are commonly used [36]. While the advantage of intramuscular vaccine administration lies in injecting exact dosages and inducing high serum IgG levels [37,38], intranasal respiratory mucosal vaccines deliver a key benefit in that mucosal surfaces, which are directly exposed to the targeted pathogens, can induce more direct and comprehensive immune responses by more closely mimicking the natural infection route compared with intramuscular vaccines [17,18,39,40]. Additionally, owing to the presence of SARS-CoV-2 spike-specific IgA in organs where SARS-CoV-2 infection occurs, intranasally administered vaccines result in faster initial clearance of SARS-CoV-2 than that of intramuscular vaccines, leading to lower viral titers in respiratory organs [41,42,43]. Apart from its immunological benefits, intranasal vaccination does not require specialized personnel for administration using syringes, allowing rapid and large-scale vaccination with minimal discomfort [44]. Furthermore, other complications that can potentially follow intramuscular injections can be avoided when vaccines are inadvertently injected into blood vessels [45].

K18-hACE2 TG mice, originally developed for SARS-CoV-1 studies, are well-established animal models that can produce mild to lethal COVID-19 depending on the infection dose or virus strain [46,47]. In addition to clear disease manifestations, K18-hACE2 mice are susceptible to upper and lower respiratory infections causing anosmia and severe pneumonia [48]. For these reasons, we used K18-hACE2 mice to evaluate the efficacy of the intranasal rK148/beta-S vaccination against lethal SARS-CoV-2 infection. Intranasal administration yielded comparable amounts of serum antibodies against both the vector virus (rK148) and SARS-CoV-2 compared with our previous study on intramuscular vaccination with the same rK148/beta-S vaccine [32]. Although neutralizing antibodies against SARS-CoV-2 were not directly observed in this study and surrogate ELISA was used, there is a correlation between the commercial ELISA-based surrogate neutralization tests and the whole virus neutralization test [49,50]. Surprisingly, immune responses to the SARS-CoV-2 spike antigen were similar in splenic immune cells after both intranasal and intramuscular vaccinations. These findings suggest that the intranasal administration of rK148/beta-S is as effective as intramuscular vaccination for stimulating systemic immunity in K18-hACE2 mice. Similar observations were reported for a virus-like particle (VLP) vaccine administered intranasally [51]. This vaccine induced comparable levels of IgG in the blood and lungs as intramuscular vaccines, along with promotion of IgA production in the blood, lungs, and upper respiratory tract (nasal wash). Since the primary receptor for NDV attachment is the ubiquitous sialic acid-containing surface glycoprotein [52], it can be inferred that rK148/beta-S exposed to respiratory cells not only stimulates the local immune response but also induces the systemic response to some extent without causing weight gain or respiratory complications. 

Various intranasal vaccines have been developed to prevent SARS-CoV-2 infection. Representative examples include live-attenuated adenoviral vectors, subunit vaccines mixed with adjuvants, MVA vectors, and parainfluenza vector vaccines [41,51,53,54,55], as follows: (1) Adenoviral vector SARS-CoV-2 vaccine: six weeks after the first vaccination, SARS-CoV-2-specific IgG and IgA were detected in the blood. Following SARS-CoV-2 challenge, the virus was not detected in the lungs and nasal wash [41]; (2) SARS-CoV-2 VLP vaccine: after the first dose, a booster dose is necessary for the generation of SARS-CoV-2-specific IgG. However, in the SARS-CoV-2 challenge test, intranasal administration showed a higher survival rate than that of intramuscular administration. Furthermore, primary intramuscular injection and booster intranasal vaccination demonstrated superior performance in all metrics [51]; (3) MVA vector SARS-CoV-2 vaccine: SARS-CoV-2-specific IgG was detected in the blood in the third week after the first vaccination, but could not completely prevent viral shedding from the upper respiratory tract. However, after the second vaccination, viral shedding from the upper respiratory tract was completely inhibited. In this experiment, intranasal–intranasal was similar to that of intramuscular–intranasal in all parameters [53]; and (4) Parainfluenza vector SARS-CoV-2 vaccine: SARS-CoV-2-specific IgG was detected in the blood in the fourth week after the first vaccination, and even after challenge with SARS-CoV-2, viral shedding was not observed from the lungs or brain compared with SARS-CoV-2 inactivated vaccine [54]. In this study, we used NDV vector platform as an intranasal vaccine against SARS-CoV-2. Similar to a previous study using viral vectors expressing SARS-CoV-2 antigen, SARS-CoV-2-specific IgG was detected in the blood in the fourth week after the first vaccination. By the fourth week after the second vaccination, higher levels and more uniformly distributed antibodies were observed. After the initial high-dose vaccination, some mice showed low antibody titers against SARS-CoV-2 and NDV. These mice were more likely to survive the SARS-CoV-2 challenge and exhibited a slight reduction in viral shedding from the respiratory tract. However, mice that received the second vaccination exhibited consistent antibody titers and demonstrated complete protection against viral shedding from the respiratory tract by day 5 dpc [56]. Although we were unable to use more recent SARS-CoV-2 variants for the challenge test, Warner et al. showed that NDV vector SARS-CoV-2 vaccine based on the Wuhan strain protected against the Omicron variant, with no viral load detected in the nasal turbinates and lungs [56].

Histopathological analysis showed that the loss of villi in the small intestine was more pronounced in the non-vaccinated group than in the vaccinated group after challenge with SARS-CoV-2. In other studies, it has been evident that the loss of villi in the small intestine is less severe in the vaccinated group compared with negative control groups [57,58]. In the lungs, perivascular inflammatory cells primarily respond to infection, damage, and other stimuli by activating immune cells and promoting immune responses against external pathogens. This response helps minimize respiratory tissue damage, supports tissue recovery, and repairs in response to infections or injuries, as perivascular inflammatory cells release inflammatory mediators that accumulate immune cells and regulate inflammatory responses in tissues, thereby aiding the proper functioning of the immune system [59,60,61]. In other studies, mouse models infected with SARS-CoV-2 showed a high level of perivascular inflammation, which was comparatively low in the vaccinated group [57,62]. In contrast, rK148/beta-S-vaccinated mice showed a slight increase in perivascular inflammation after SARS-CoV-2 inoculation. However, this unexpected inflammatory response might be attributed to the specificity of the SARS-CoV-2 lethal model in K18-hACE2 mice, which exhibit high ACE2 expression in the lung cells, and to the abnormally high titers of challenge virus required to induce lethality. Additionally, given that a 100 µL dose in the nasal cavity reaches most parts of the lungs [63], a pathological evaluation and assessment of intranasal administration should be confirmed using a larger animal model that allows for nasal spray delivery limited to the nasal cavity [64]. Also, the intranasal delivery of vaccines can directly reach the brain, necessitating further research on safety aspects [40].

In contrast to other NDV vector SARS-CoV-2 vaccines, rK148/beta-S was not developed using the original Wuhan strain of SARS-CoV-2 [65]. The SARS-CoV-2 Beta variant has significantly increased its ability to evade immunity from existing vaccines due to the E484K mutation in the spike protein [66]. The rK148/beta-S vaccine was designed to target the Beta variant of SARS-CoV-2, which has the E484K mutation in the spike protein. However, the Beta variant did not become the dominant strain, with the Delta variant emerging as the predominant one [67]. Despite this, rK148/beta-S also demonstrated excellent protective efficacy against the Delta variant. rK148/beta-S did not enhance the stability of the spike protein using hexa-pro. Nonetheless, it demonstrated sufficient efficacy as a vaccine. This suggests the potential for rapidly developing vaccines for emergent respiratory diseases by expressing the original glycoprotein. The backbone of the NDV vector vaccine used in rK148/beta-S is derived from the K148/08 strain, which originated from ducks. Compared to the commonly used LaSota strain, which has a long history as a commercial NDV vaccine, the K148/08 strain represents a relatively recent development genetically close to emerging NDV [23]. This suggests that the potential for expanding the range of NDV vector vaccines for human use extends beyond traditional strains to include other APMV-1 (NDV) strains. However, it is crucial to thoroughly validate the safety profile of these newer strains in clinical trials.

In this study, the intranasal administration of live rK148/beta-S resulted in 100% survival rate and significantly reduced the lung viral load after lethal COVID-19 infection. Additionally, intranasal administration of live rK148/beta-S induced SARS-CoV-2 spike-specific IgA in the lungs. rK148/beta-S is a potential vaccine candidate for SARS-CoV-2 infection. Further safety testing and thorough evaluations are warranted to explore its usability.

## Figures and Tables

**Table 1 vaccines-12-00921-t001:** Design of the rK148/beta-S intranasally vaccinated and challenge schedule.

Group	Immunization				Challenge		
	Vaccine	Dose (EID_50_)	Prime	Boost	Virus	Dose (TCID_50_)	Time Point
**G1**	rK148/beta-S	10^6.0^	W0	W4	Beta ^a^	10^4.7^	W8
**G2**	rK148/beta-S	10^7.0^	W0	W4	Beta	10^4.7^	W8
**G3**	NDV (K148/08)	10^7.0^	W0	W4	Beta	10^4.7^	W8
**G4**	Control (PBS)	–	—	—	Beta	10^4.7^	W8
**G5**	rK148/beta-S	10^7.0^	W0	W4	Delta ^b^	10^4.7^	W8
**G6**	Control (PBS)	–	—	—	Delta	10^4.7^	W8

^a^ SARS-CoV-2 Beta variant; ^b^ SARS-CoV-2 Delta variant.

## Data Availability

The data presented in this study are available on request from the corresponding author.

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
