# Peer review of "Intranasal Administration of Recombinant Newcastle Disease Virus Expressing SARS-CoV-2 Spike Protein Protects hACE2 TG Mice against Lethal SARS-CoV-2 Infection"

_vaccines, 2024, doi:10.3390/vaccines12080921_

Round 1

Reviewer 1 Report

Comments and Suggestions for Authors

Comments to the author

This research documented that the Newcastle disease virus SARS-CoV-2 rK148/beta-S intranasal immunization is efficacious and may produce mucosal immunity against SARS-CoV-2. The article is well-organized and contains novel concepts. I'd like to provide the following suggestions:

1. Line 108, Please rewrite EID50 so that it is not unclear.

2. Please describe the RT-PCR target and internal reference.

Author Response

we are grateful to the reviewers for the valuable suggestions provided. Through this major revision, the paper has become more sophisticated.

Here are responses to the reviewer comments: This research documented that the Newcastle disease virus SARS-CoV-2 rK148/beta-S intranasal immunization is efficacious and may produce mucosal immunity against SARS-CoV-2. The article is well-organized and contains novel concepts. I'd like to provide the following suggestions:

1. Line 108, Please rewrite EID50 so that it is not unclear.
Answer: It was re-written for clear and added to line 114~116. “Each vaccine was administered intranasally with 100 μL of live virus at a concentration of either 107.0 EID50/mL (low dose) or 108.0 EID50/mL (high dose)”

  1. Please describe the RT-PCR target and internal reference.
    Answer: Thank you very much for your comments. In line 180, it is described that the target is the NDV M gene. We used the same primers and probes as those described in the reference 34, "Development of a Real-Time Reverse-Transcription PCR for Detection of Newcastle Disease Virus RNA in Clinical Samples" for this experiment. Given the widespread use of this primer for NDV detection, we consider it unnecessary to reiterate the method.

we would be happy to make any further changes that may be required.

Reviewer 2 Report

Comments and Suggestions for Authors

The authors assess an intranasal NDV vectored SARS-CoV-2 spike vaccine in an hACE-2 RG mouse model.

It is important to develop mucosal vaccines especially against respiratory infections. The NDV vector is an appropriate system to develop mucosal vaccines.

The experimental methods are sound and the results of the study are characterized soundly.

Line 153 CO2

The Surrogate SARS-CoV-2 ELISA demonstrates there are antibodies against SARS-CoV-2. However with this assay it is difficult to see differences in the quality of the antibody responses. Do you have SARS-CoV-2 neutralizing antibody responses which could be used to compare the antibody responses between the 2 timepoints and doses of rk148/beta-S vaccine? There seems to be know statistical difference between the 2 doses, however clinically there are differences.

In addition, some mice are responding following a single vaccination.  What is the potential level of protection if mice were challenged after a single vaccination?

In the discussion can the authors discuss the translation of the results from mice into other host species.

Author Response

We are grateful to the reviewers for the valuable suggestions provided. Through this major revision, the paper has become more sophisticated.

Here are responses to the reviewer comments: The authors assess an intranasal NDV vectored SARS-CoV-2 spike vaccine in an hACE-2 RG mouse model. It is important to develop mucosal vaccines especially against respiratory infections. The NDV vector is an appropriate system to develop mucosal vaccines. The experimental methods are sound and the results of the study are characterized soundly.

  1. Line 153 CO2

Answer: Now it is corrected with the subscript, CO2 (line 163)

  1. The Surrogate SARS-CoV-2 ELISA demonstrates there are antibodies against SARS-CoV-2. However with this assay it is difficult to see differences in the quality of the antibody responses. Do you have SARS-CoV-2 neutralizing antibody responses which could be used to compare the antibody responses between the 2 timepoints and doses of rk148/beta-S vaccine? There seems to be know statistical difference between the 2 doses, however clinically there are differences.

Answer: Thank you very much for your comments. we agree with this comment. Instead of the whole virus neutralization test, which can only be performed in BSL-3 conditions, we focused on evaluating antibody levels using Commercial ELISA-Based Surrogate Neutralization Tests. These tests are simpler, do not require cells, and can be conducted in BSL-2 conditions. Zedan et al., “Evaluation of commercially available fully automated and ELISA‑based assays for detecting anti‑SARS‑CoV‑2 neutralizing antibodies” and Silva et al., “Validation of a SARS-CoV-2 Surrogate Neutralization Test Detecting Neutralizing Antibodies against the Major Variants of Concern” demonstrated the correlation between the Commercial ELISA-Based surrogate neutralization tests and the whole virus neutralization test. Although an exact match between the two methods was not observed, a significant correlation trend was evident. We have added those references in the discussion section, lines 434~437 ” Although neutralizing antibodies against SARS-CoV-2 were not directly observed in this study and surrogate ELISA was used, there is a correlation between the commercial ELI-SA-Based surrogate neutralization tests and the whole virus neutralization test”, addressing the correlation between our surrogate ELISA and neutralization. If you are referring to other features related to antibody quality, such as duration, affinity, or isotype switching, we regret that limited resources and funding prevented us from conducting further research. We are eager to elucidate a more in-depth immune response, focusing on those features, in a follow-up study.

  1. In addition, some mice are responding following a single vaccination. What is the potential level of protection if mice were challenged after a single vaccination?

Answer: Thank you very much for your comments. we agree with this comment. we add this comment to line 472~477. “After the initial high-dose vaccination, some mouse showed low antibody titers against SARS-CoV-2 and NDV. These mice were more likely to survive SARS-CoV-2 challenge and exhibited a slight reduction in viral shedding from the respiratory tract. However, mice that received the second vaccination exhibited consistent antibody titers and demonstrated complete protection against viral shedding from the respiratory tract by day 5 dpc”

  1. In the discussion can the authors discuss the translation of the results from mice into other host species.

Thank you very much for your comments. we agree with this comment. we add this comment to line 79~82. “The NDV vector used as the platform for the vaccine: 1) SARS-CoV-2; SARS-CoV-2 vaccine is also applicable for use in pigs, and clinical trials in humans have already been completed, 2) Bovine ephemeral fever virus vaccine for used bovine, 3) Rabies virus vaccine for used canines and felines”

we would be happy to make any further changes that may be required.

Reviewer 3 Report

Comments and Suggestions for Authors

In this report the authors use a NVD vectored vaccine against SARSCoV2 to proof the concept that nasal immunization induces an effective immune response, which was proven in a mice model. Antibody and cellular responses were measured and survival rates in mice challenged with SARSCpV2

The following suggestions are identified to improve the manuscript:

1.      The argument for the need of mucosal vaccines can be improved by highlighting the fact that current injectable vaccines effectively prevent severe COVID-19 but are of low efficacy in the prevention of the infection and transmission among individuals.

2.      In the methods section, it is advisable to provide more details on the immunization protocol (anesthesia was used? The vaccine was administered using a micropipette? For how long the animals were immobilized?).

3.      The authors should discuss in more detail what their study adds to the field when compared with the reports from Warner et al and others (i.e. doi: 10.1101/2022.02.08.22270676. and doi: 10.1128/mBio.01908-21.).

4.      Although the authors highlight the advantages of nasal immunization, is there any disadvantage on this approach? (i.e. difficulties for a homogeneous dosage due to the anatomy and the administration methods for this type of vaccines).

Author Response

Reviewer 3

we are grateful to you and the reviewers for the valuable suggestions provided. Through this major revision, the paper has become more sophisticated.

Here are responses to the reviewer comments: In this report the authors use a NVD vectored vaccine against SARSCoV2 to proof the concept that nasal immunization induces an effective immune response, which was proven in a mice model. Antibody and cellular responses were measured and survival rates in mice challenged with SARS-CoV-2. The following suggestions are identified to improve the manuscript:

  1. The argument for the need of mucosal vaccines can be improved by highlighting the fact that current injectable vaccines effectively prevent severe COVID-19 but are of low efficacy in the prevention of the infection and transmission among individuals.

Answer: Thank you very much for your comments. We agree with this comment. we add this comment to line 63~65. “Although current intramuscular vaccines are highly effective in preventing severe COVID-19, their efficacy in preventing infection and transmission among individuals re-mains low”

  1. In the methods section, it is advisable to provide more details on the immunization protocol (anesthesia was used? The vaccine was administered using a micropipette? For how long the animals were immobilized?).

Answer: Thank you very much for your comments. we agree with this comment. we add this comment to line 113~114. “Intranasal administration of the vaccine or challenge virus was conducted by instillation into the nostrils following anesthesia”

  1. The authors should discuss in more detail what their study adds to the field when compared with the reports from Warner et al and others (i.e. doi: 10.1101/2022.02.08.22270676. and doi: 10.1128/mBio.01908-21.).

Answer: Thank you very much for your comments. we agree with this comment. we add this comment to line 503~520. “In contrast to other NDV vector SARS-CoV-2 vaccines, rK148/beta-S was not devel-oped using the original Wuhan strain of SARS-CoV-2. The SARS-CoV-2 beta variant has significantly increased its ability to evade immunity from existing vaccines due to the E484K mutation in the spike protein [64]. The rK148/beta-S vaccine was designed to target the beta variant of SARS-CoV-2, which has the E484K mutation in the spike protein. However, the beta variant did not become the dominant strain, with the delta variant emerging as the predominant one. Despite this, rK148/beta-S demonstrated excellent protective efficacy against the delta variant as well. rK148/beta-S did not enhance the sta-bility of the spike protein using hexa-pro. Nonetheless, it demonstrated sufficient efficacy as a vaccine. This suggests the potential for rapidly developing vaccines for emergent res-piratory diseases by expressing the original glycoprotein. The backbone of the NDV vector vaccine used in rK148/beta-S is derived from the K148/08 strain, which originated from ducks. Compared to the commonly used LaSota strain, which has a long history as a commercial NDV vaccine, the K148/08 strain represents a relatively recent development genetically close to emerging NDV. This suggests that the potential for expanding the range of NDV vector vaccines for human use beyond traditional strains to include other APMV-1 (NDV) strains. However, it is crucial to thoroughly validate the safety profile of these newer strains in clinical trials.”

  1. Although the authors highlight the advantages of nasal immunization, is there any disadvantage on this approach? (i.e. difficulties for a homogeneous dosage due to the anatomy and the administration methods for this type of vaccines).

Thank you very much for the insightful comments. In line 500~502, it is stated that further experiments are needed to determine the appropriate dosage for nasal vaccination, as well as the amount reaching the target in larger animals, depending on the device used. Additionally, since the nasal cavity is near the brain, potential risks associated with this route are discussed in line 498. "Also, the intranasal delivery of vaccines can directly reach the brain, necessitating further research on safety aspects"

we would be happy to make any further changes that may be required
